# Frequency of Peripheral CD8+ T Cells Expressing Chemo-Attractant Receptors CCR1, 4 and 5 Increases in NPC Patients with EBV Clearance upon Radiotherapy

**DOI:** 10.3390/cancers15061887

**Published:** 2023-03-21

**Authors:** Shweta Mahajan, Hayri E. Balcioglu, Astrid Oostvogels, Willem A. Dik, K. C. Allen Chan, Kwok-Wai Lo, Edwin P. Hui, Anna Tsang, Joanna Tong, Wai Kei Jacky Lam, Kenneth Wong, Anthony T. C. Chan, Brigette B. Y. Ma, Reno Debets

**Affiliations:** 1Laboratory of Tumor Immunology, Department of Medical Oncology, Erasmus MC Cancer Institute, 3015 GD Rotterdam, The Netherlands; 2Laboratory of Medical Immunology, Department of Immunology, Erasmus MC, 3015 GD Rotterdam, The Netherlands; 3Department of Chemical Pathology, The Chinese University of Hong Kong, Hong Kong SAR, China; 4Li Ka Shing Institute of Health Sciences, The Chinese University of Hong Kong, Hong Kong SAR, China; 5Department of Anatomical and Cellular Pathology, The Chinese University of Hong Kong, Hong Kong SAR, China; 6State Key Laboratory of Translational Oncology, Sir YK Pao Centre for Cancer, Department of Clinical Oncology, Hong Kong Cancer Institute, The Chinese University of Hong Kong, Hong Kong SAR, China

**Keywords:** Epstein–Barr virus, nasopharyngeal carcinoma, T-cell subsets, blood, chemoattractant receptors

## Abstract

**Simple Summary:**

The post-radiotherapy (RT) clearance of tumor-derived plasma EBV DNA is associated with a better prognosis in patients with nasopharyngeal cancer (NPC). T cells may play a critical role in the clearance of plasma EBV DNA in NPC. We explored the dynamic changes in circulating T-cell profiles during RT and observed that a temporal increase in the frequency of CD8+ T cells expressing CCR1, 4 and/or 5 was associated with plasma EBV DNA clearance. Moreover, differences in the plasma levels of the corresponding chemo-attractants are related to clinical outcome. Our study demonstrated correlation between plasma EBV DNA clearance post-RT and T-cell chemotaxis, which requires validation in larger cohorts. Markers of T-cell chemotaxis may be prognostic biomarkers in patients with NPC undergoing RT.

**Abstract:**

Radiotherapy (RT) is the standard-of-care for Epstein–Barr virus (EBV)-associated nasopharyngeal carcinoma (NPC), where the post-RT clearance of plasma EBV DNA is prognostic. Currently, it is not known whether the post-RT clearance of plasma EBV DNA is related to the presence of circulating T-cell subsets. Blood samples from NPC patients were used to assess the frequency of T-cell subsets relating to differentiation, co-signaling and chemotaxis. Patients with undetectable versus detectable plasma EBV DNA levels post-RT were categorized as clearers vs. non-clearers. Clearers had a lower frequency of PD1+CD8+ T cells as well as CXCR3+CD8+ T cells during RT compared to non-clearers. Clearers exclusively showed a temporal increase in chemo-attractant receptors CCR1, 4 and/or 5, expressing CD8+ T cells upon RT. The increase in CCR-expressing CD8+ T cells was accompanied by a drop in naïve CD8+ T cells and an increase in OX40+CD8+ T cells. Upon stratifying these patients based on clinical outcome, the dynamics of CCR-expressing CD8+ T cells were in concordance with the non-recurrence of NPC. In a second cohort, non-recurrence associated with higher quantities of circulating CCL14 and CCL15. Collectively, our findings relate plasma EBV DNA clearance post-RT to T-cell chemotaxis, which requires validation in larger cohorts.

## 1. Introduction

Nasopharyngeal cancer is a head and neck cancer with a unique geographical distribution and is endemic to southern China, southeast Asia and North Africa. The endemic form of NPC is of a non-keratinizing histological subtype rich in lymphocytic infiltrate and shows the presence of Epstein–Barr virus (EBV) infection in tumor cells [1,2,3]. Patients with non-metastatic NPC are primarily treated with radiotherapy (RT) with or without chemotherapy depending on the stage of the disease [4,5]. With modern RT, loco-regional recurrence occurs in 10–15% of patients [2,3]. Quantifiable biomarkers in combination with clinical staging in NPC would improve prognostic stratification, treatment allocation and post-treatment monitoring [6]. Currently, the EBV DNA level in plasma, quantified by real-time polymerase chain reaction (RT-PCR), is the most widely used biomarker in patients with NPC. Plasma EBV DNA accurately reflects tumor burden in patents with NPC and has a predictable half-life following RT or surgery in NPC [7,8]. In fact, a meta-analysis of 40 studies concluded that detectable plasma EBV DNA levels during or post-RT are associated with a worse clinical outcome [9]. At present, any mechanistic connection between plasma EBV DNA clearance and host immunity remains to be defined. 

Intra-tumoral CD8+ T cells are an important component of the NPC tumor immune microenvironment and contribute to anti-NPC response as reported in a meta-analysis by Berele and colleagues [10]. In addition, many sequencing studies at exome, genome or single-cell level in primary and recurrent NPC have revealed T-cell phenotype(s) as distinct biomarker(s) for prognosis and treatment response [11,12,13,14]. For example, a study by Liu and colleagues identified an immune signature (PD-L1+, CD163+ and expression levels of CXCR5 and CD117) with prognostic value for progression-free survival [15]. In another single-cell RNA sequencing (scRNA) study of 14 tissue samples with either NPC or nasopharyngeal lymphatic hyperplasia (NLH), a higher frequency was found of T cells expressing the exhaustion-related gene *HAVCR2* in NPC tumors compared to NLH tissues [13]. These studies strongly suggest that EBV-associated NPC employ several CD8+ T-cell-evasive strategies [11,13,16,17]. Along this line, EBV-related gene products expressed by NPC are recognized for their immunosuppressive actions against CD8+ T cell numbers and function. For instance, in vitro analysis by Cai and colleagues demonstrated that LMP1 promoted the recruitment of myeloid-derived suppressor cells (MDSCs) via the release of IL-1ß, IL-6 and GM-CSF [18,19]. Another in vitro study by Huo and colleagues demonstrated that the EBNA1 protein was actively involved in the recruitment of regulatory T cells (Treg) via the up-regulated expression of TGF-ß [20]. A study by Tao and colleagues investigated the association between circulating immune cell frequencies, EBV viral loads and clinical outcome and demonstrated a significant association between plasma EBV DNA positivity and CD8+ T-cell presence [21].

The exact and dynamic changes in T-cell subsets during RT in relation to plasma EBV DNA clearance are relatively unknown in NPC patients. A quest for such data would aid our understanding of NPC and potentially provide novel markers (and targets) for better patient stratification and therapy development. The profiling of various receptors expressed by T cells according to maturation, co-stimulation and co-inhibition as well as chemotaxis in serial blood samples of NPC patients who received RT could help define the status and function of T cells and potentially provide a reflection of the fitness of the anti-tumor immune response. Currently there are no quantifiable circulating markers for NPC other than post-RT plasma EBV DNA clearance, for which reason we have assessed T-cell phenotypes as a function of this clearance.

Here, we have conducted a prospective pilot study in which we have profiled circulating T-cell subsets of NPC patients who did versus those who did not achieve plasma EBV DNA clearance post-RT. Our findings indicate treatment-induced temporal changes and notable differences in the abundance of specific T-cell subsets in the context of post-RT plasma EBV DNA clearance and clinical outcome. These findings indicate that defined T-cell subsets, particularly those related to chemotaxis, are associated with plasma EBV DNA clearance and delineate potential markers and targets for immunotherapies for NPC patients.

## 2. Materials and Methods

### 2.1. Patient Eligibility and Collection of Biomaterials

Biospecimens from 2 cohorts of patients were included in this study: prospectively collected patient materials (Cohort 1, n = 24) and retrospectively archived patient materials (Cohort 2, n = 28). Patients of Cohort 1 with a histologically confirmed diagnosis of Stage I to IVA undifferentiated NPC (American Joint Committee of Cancer, AJCC, 8th edition) and who were scheduled for RT at the Prince of Wales Hospital (Hong Kong SAR, China) were prospectively enrolled for the current study between January 2018 and October 2020 [4]. All patients underwent staging with contrast-enhanced MRI of the nasopharynx and neck as well as systemic imaging with contrast-enhanced CT or PET scan and underwent intensity-modulated RT (IMRT) at 70 Gy in 33–35 fractions over 7 weeks. In addition, patients with Stage II to IVA were treated with concurrent cisplatin (40 mg/m^2^, weekly given intravenously) during IMRT, with or without prior induction chemotherapy. Peripheral blood was collected in EDTA tubes at 3 time points: pre-radiotherapy (pre-RT), at week 4 of RT (on-RT) and between 4 and 6 weeks after RT (post-RT) (see Figure 1A for study design). Nasopharyngeal cancer recurrence is confirmed either by radiologic imaging (MRI, CT or PET) or a naso-endoscopy with/without biopsy. Five milliliters of EDTA blood was used to collect plasma for EBV DNA quantification or detection of chemo-attractants, and fifteen milliliters of EDTA blood was used to isolate peripheral blood mononuclear cells (PBMCs). The latter isolation was performed using ficoll gradient centrifugation, and isolated PBMCs were aliquoted and stored in liquid nitrogen. Study participants gave written informed consent prior to enrollment, and the study protocol was approved by the Joint Chinese University of Hong Kong–New Territories East Cluster Clinical Research Ethics Committee, Hong Kong SAR (CREC study number 2016.439, protocol 12.5.21 version 4). From patients of Cohort 2, plasma was sampled before RT (as described for Cohort 1). Characteristics of patients of both cohorts are outlined in Table 1.

### 2.2. Definition of Plasma EBV DNA Clearance

Plasma EBV DNA titers were measured for Cohort 1 using quantitative real-time PCR targeting the *BamH*I-W gene of EBV as previously published [22]. The lower limit of detection of this assay is 20 copies/mL. Patients with undetectable plasma EBV DNA (less than 20 copies/mL) or detectable (at least 20 copies/mL) plasma that was collected post-RT were categorized as clearers or non-clearers, respectively.

### 2.3. Flow Cytometry Analysis

The PBMCs of Cohort 1 were thawed in a batchwise manner (with all timepoints of each patient in a single batch), stained with panels of antibodies targeting >20 markers of T-cell maturation, co-inhibition, co-stimulation and chemotaxis and measured on the BD 3-laser Celesta flow cytometer using the FACSDIVA 8.x software. Panels of antibodies have been optimized and tested before, and the data were analyzed using Flowjo software [23,24].

### 2.4. T-Cell Subset Clustering and Uniform Manifold Approximation and Projections

Analysis was carried out for Cohort 1 using an in-house written python (v3.6.8) script as previously described [23,24]. In brief, CD4 and CD8 T cells obtained from FlowJo were corrected for spillover and background, logicle transformed and normalized. These data were then subjected to clustering with a self-organizing map algorithm with 100 nodes, mapping to 20 meta-clusters and further cluster merging based on marker expressions in clusters so each cluster corresponded to a unique set of markers. This resulted in 16 to 20 unique clusters per T-cell panel. The differential abundance of clusters was analyzed between patient groups (clearers and non-clearers) and timepoints (pre-RT, on-RT and post-RT) and uniform manifold approximations and projections (UMAPs) were generated for their visualization.

### 2.5. Quantification of Chemo Attractants

The chemo-attractants CCL2, CCL3, CCL4, CCL5, CCL8, CCL11, CCL13, CCL14, CL15, CCL17, CCL22, CCL23, CXCL9, CXCL10, CXCL11 and CXCL13 were measured in plasma samples of Cohort 2 using the Luminex assay (R&D systems, Abingdon, UK), which was performed according to the manufacturer’s recommendations and measured on the Luminex Magpix machine.

### 2.6. Statistical Analysis

Differences regarding T-cell phenotypes in blood or chemo-attractant levels in plasma between clearers and non-clearers were assessed using the Mann–Whitney U test, whereas differences between pre-RT, on-RT and post-RT were assessed using the paired Wilcoxon signed rank test. Correlations were tested using Spearman analysis. Differences with *p* value < 0.05 were considered statistically significant.

## 3. Results

### 3.1. Patient and Study Characteristics

Cohort 1 patients were 83% male with an overall median age of 51 years and 75% of patients had Stage III-IVA NPC. These patients were treated with RT or concurrent Chemo-Radiation Therapy (CRT), and 20% of patients received induction chemotherapy (Table 1). After median follow-up of 2 years, patients were classified as ‘plasma EBV DNA clearer’ (n = 18) or ‘non-clearer’ (n = 6) based on whether they had undetectable or detectable plasma EBV-DNA 4–6 weeks after RT (Figure 1A,B). At the time of cut-off, four patients in Cohort 1 were confirmed as recurring and all of them were classified as non-clearers. Cohort 2 patients were 75% male with an overall median age of 55 years and 57% of patients had Stage III-IVA NPC. These patients were also treated with (C)RT, and 25% received induction chemotherapy (Table 1). Following a median follow-up of 6 years, 25% of Cohort 2 patients were confirmed to show recurrence.

### 3.2. Nasopharyngeal Cancer Patients with Plasma EBV DNA Clearance Exhibit Lower Abundance of Circulating PD1+CD8+ T Cells On- and Post-RT When Compared to Patients without Plasma EBV DNA Clearance

Regarding T-cell phenotyping, we first analyzed the frequencies of CD8+ T cells expressing the following co-inhibitory receptors: programmed cell death protein (PD1) (CD279); T-cell immunoglobulin and mucin-domain-containing-3 (TIM3) (CD366); B- and T-lymphocyte attenuator (BTLA) (CD272); and/or LAG3 (CD223) in Cohort 1 patients. The unsupervised clustering of CD8 T cells resulted in 20 distinct clusters, 17 of which showed differential abundance when comparing the two patient subgroups at each timepoint during treatment, or when comparing the three treatment timepoints for a single patient group. Specifically, we observed that clearers versus non-clearers had a lower abundance of clusters comprising CD8+ T cells that express PD1 on- or post-RT (clusters 9 and 12) (Figure 2A–C). We extended the dimensionality reduction analysis with the analysis of defined subsets of CD8+ T cells expressing single or two types of co-inhibitory receptors. The latter analyses confirmed a lower abundance of PD1+CD8+ T cells on- and post-RT as well as a lower abundance of PD1+LAG3+CD8+ T cells post-RT in plasma EBV DNA clearers compared with non-clearers (Figure 2D,I). Moreover, during RT we observed an increase in the abundance of TIM3+, PD1+TIM3+ and LAG3+BTLA+CD8+ T cells (i.e., difference among timepoints) in plasma EBV DNA clearers but not non-clearers (Figure 2E,J,M). Interestingly, we also observed a decrease in the abundance of BTLA+CD8+ T cells and an increase in the abundance of LAG3+ and TIM3+LAG3+CD8+ T cells during RT in plasma EBV DNA clearers as well as non-clearers (Figure 2F,G,L).

Next, we investigated the differentiation status of CD8+ T cells according to the expression of C-C Chemokine receptors 7 (CCR)7 and CD45RA. During RT, we observed an early and steep drop in the frequency of naïve CD8+ T cells as well as an increase in the abundance of effector memory CD8+ T cells in plasma EBV DNA clearers but not non-clearers (Figure 3A,C).

### 3.3. Nasopharyngeal Cancer Patients with Plasma EBV DNA Clearance Had a Late Rise in the Abundance of OX40+ CD8+ T Cells during RT

In addition to co-inhibitory receptors, we also analyzed the frequencies of CD8+ T cells expressing the following co-stimulatory receptors: CD28; Inducible T-cell COStimulator (ICOS) (CD278); CD40L (CD154); 41BB (CD137); and/or OX40 (CD134) in Cohort 1 patients (Figure 4 and Appendix A). Plasma EBV DNA clearers, but not non-clearers, showed an increase in the abundance of OX40+, OX40+ICOS+ and OX40+CD40L+CD8+ T cells at the post-RT time point (Figure 4E,J,K). Interestingly, we observed a decrease in the abundance of CD28+CD40L+CD8+ T cells during RT in both patient subgroups regardless of the plasma EBV clearance status (Figure 4G).

### 3.4. Nasopharyngeal Cancer Patients with Plasma EBV DNA Clearance Demonstrate an Early and Profound Rise in the Abundance of CCR1+, CCR4+ and/or CCR5+CD8+ T Cells during RT

We analyzed the frequencies of CD8+ T cells expressing the following chemo-attractant receptors: CCR1 (CD191); CCR4 (CD194); CCR5 (CD195); CXCR3 (CD183); and/or CXCR4 (CD184) in Cohort 1 patients. (Figure 5 and Appendix A). Regarding these chemo-attractant receptors, we observed two differences between clearers and non-clearers. First, plasma EBV-DNA clearers had a lower abundance of CXCR3+ and CXCR3+CXCR4+CD8+ T cells on-RT when compared to non-clearance (Figure 5D,O). Second, plasma EBV DNA clearers demonstrated a sharp and early increase in the frequencies of CD8+ T cells expressing CCR1, 4 and/or 5, which was also observed for CD8+ T cells co-expressing one of these CCRs and CXCR4 (Figure 5A–C,F,J,M).

### 3.5. Abundance of CCR1, CCR4 and/or CCR5+CD8+ T Cells Is Associated with Presence of Mature OX40+CD8+ T Cells

Our findings regarding the profiling of circulating CD8+ T cells in the blood of NPC patients in relation to plasma EBV DNA clearance are summarized in Appendix A. In an effort to relate the findings of Cohort 1 patients regarding chemo-attractant receptors to those related to T-cell co-inhibition, co-stimulation and T-cell differentiation, we performed correlative analyses. As displayed in Figure 6, a lowered frequency of CXCR3-expressing CD8+ T cells is associated with a lowered frequency of PD1-expressing CD8+ T cells. Next to that, an increase in the frequency of CD8+ T cells expressing CCR1, CCR4 and/or CCR5 is associated with an increase in the frequency of both more mature CD8+ T cells (CCR7^-^CD45RA^-^) and 0X40+ CD8+ T cells.

We also analyzed the immune profile of CD4+ T cells in the blood of the NPC patients of Cohort 1. Interestingly, we observed a highly similar profile to that of CD8+ T cells (see for overview Appendix A). For instance, similar to CD8+ T cells during RT, plasma EBV DNA clearers but not non-clearers demonstrated a late rise in the abundance of PD1+TIM3+CD4+ T cells (Appendix A). In line with the similarities between CD8+ and CD4+ T-cell profiles, plasma EBV DNA clearers also demonstrated an early and steep drop in the abundance of naïve CD4+ T cells, which was accompanied by an increase in the abundance of central and effector memory CD4+ T cells (Appendix A); also, this patient subgroup demonstrated an early rise in the abundance of OX40+CD4+ T cells (Appendix A). Lastly, this subgroup also uniquely demonstrated a treatment-induced increase in the abundance of CCR1+, CCR1+CXCR4+, CCR4+CXCR4+ CD4+ T cells (Appendix A).

### 3.6. Patients without NPC Recurrence Showed Differential Presence of Markers of T-Cell Chemotaxis When Compared to Patients with NPC Recurrence

When re-stratifying Cohort 1 NPC patients according to clinical response, we again observed an RT-mediated increase in the abundance of CD8+ as well as CD4+ T cells expressing CCR1, CCR4 and/or CCR5 in non-recurrent NPC patients but not in recurrent patients. The same holds true for increases in the abundance of CD8+ as well as CD4+ T cells expressing OX40 or co-expressing PD1 and TIM3 (Appendix A). To further substantiate the clinical value of circulating markers of T-cell chemotaxis, we also tested whether the plasma levels of 16 putative ligands for CCR1, 4 and/or 5 were different as a function of NPC recurrence in Cohort 2. Although we did not observe any differences in CXCR3 ligands CXCL9, 10, 11 and 13, patients with no NPC recurrence had higher quantities of circulating CCL14 (ligands for CCR1 and CCR5) and CCL15 (ligand for CCR1) pre-RT when compared to patients who had NPC recurrence (Appendix A).

## 4. Discussion

In this study, we demonstrated key changes in the profile of peripheral T cells in the context of post-RT plasma EBV DNA clearance in NPC patients undergoing RT or CRT. Based on these results (Figure 2, Figure 3, Figure 4, Figure 5 and Figure 6, summary in Appendix A), we hypothesize in a schematic diagram (Figure 7) that patients who were able to clear plasma EBV DNA post-RT possess an immunologically active circulating T-cell profile with a more differentiated phenotype and enhanced chemotactic activity. This is also reflected in patients from a second cohort who did not experience disease recurrence demonstrating higher plasma CCL14 and CCL15 levels compared to patients who experienced disease recurrence. Table 2 provides a short description regarding the immunological context of the markers highlighted in this study.

The radiotherapy-induced immunogenic cell death (ICD) of tumor cells is well recognized for its effects on T-cell immunity, which are extended by the findings of the current study. For instance, ICD stimulates the recruitment of chemo-attractant receptor-expressing immune cells, modulating response to RT [36,37]. A study by Tao and colleagues demonstrated an association between CXCR4 expression in NPC tissues and clinical outcome before treatment [38]. Moreover, several scRNAseq studies have consistently reported the importance of chemo-attractant–chemo-attractant receptor interactions in modulating the infiltration of immune cells as well as the interaction between immune cells in NPC [13,39]. Similarly, in our study chemo-attractant receptors seemed to play an important role in shaping the T-cell profiles of patients with NPC when they were stratified according to their ability to clear plasma EBV DNA post-RT, and whether they developed disease recurrence post-RT. The observed increase in the abundance of CCR1, CCR4 and/or CCR5+CD8+ T cells may reflect an RT-induced T-cell response directed against NPC tissue, after which these T cells efflux from the tumor into the blood of responding patients. This suggests that CCR1, CCR4 and/or CCR5+CD8+ T cells have seen antigens and/or are triggered by chemo-attractant ligands, which would be in line with the observed higher pre-treatment levels of CCL14 and CCL15 in the plasma of non-recurrent NPC patients (Cohort 2) [40]. Similar findings were observed by Li and colleagues, who demonstrated the radiation-induced release of CCL22 by NPC PDX tumors and its role in the recruitment of CCR4+ T cells with a cytotoxic signature [41]. Kondo and colleagues have reported an activated phenotype of CCR4+CD8+ T cells, as evidenced by the production of inflammatory cytokines [42]. Our findings build further on this notion, as the frequencies of CCR1, 4 and/or 5+ T cells correlated with the frequencies of T cells with an activated phenotype, as governed by the expression of OX40 and T-cell maturation markers. Additional analysis as to whether CCR1, 4 and/or 5+ T cells produce inflammatory cytokines is relevant and is recommended for future research.

Conversely, our study pointed to a higher abundance of CXCR3+ and CXCR3+CXCR4+CD8+ T cells in plasma EBV DNA non-clearers when compared to clearers. In non-clearers, we also observed a higher abundance of PD1+ and PD1+LAG3+CD8+ T cells. In fact, the abundance of CXCR3+CD8+ T cells correlated with that of PD1+CD8+ T cells. These findings may represent a lack of T-cell infiltration and/or the presence of T-cell exhaustion in the blood of non-responding patients. The further characterization of CXCR3+PD1+ T cells using immunostaining or TCR sequencing in NPC tumors could shed light on the potential EBV genes contributing to immune evasion in NPC. This may also explain why the response rate to anti-PD1 inhibitors such as monotherapy is relatively modest in recurrent/metastatic NPC [43,44] and also why there is a lack of consistent prognostic value of PD1/PD-L1 expression for NPC patients [45]. Therefore, the combination of PD1/PDL1 with other biomarkers such as LAG3 and/or plasma EBV DNA might be clinically significant. This is supported by our study as well as by Li and colleagues, who reported that 93% of patients with detectable pre-treatment plasma EBV DNA and high PDL1 expression were associated with a worse prognosis and multiple studies reporting LAG3 expression to be enriched in dysfunctional T cells in NPC tumors [11,39,46]. Myeloid-derived suppressor cells (MDSCs) have been reported to potentially jeopardize the influx and function of T cells and may represent the underlying inducer of these exhausted T cells, which has been reported in RM-NPC patients by Hopkins and colleagues [47] and is of interest for future studies.

Another important consequence of ICD is the release of tumor-associated antigens (TAAs) and stress-associated signals that mediate immune cell proliferation and activation [37]. Interestingly, we observed that NPC patients with plasma clearance of EBV DNA demonstrated an increase in the frequencies of OX40+ CD4 as well as CD8 T cells in their blood upon RT. OX-40 is a co-stimulatory receptor that is expressed by different subsets of T cells, particularly CD4 T cells, and its expression is induced following antigen exposure [28]. Thus, the increased frequency of OX40+ T cells in the blood may be reflective of an antigen encounter in the tumor or tumor-draining lymphoid organs, the occurrence of which may have been facilitated upon the radiation-induced immunogenic cell death of cancer cells and the subsequent uptake and presentation of released antigens by myeloid cells. In fact, the interaction of T-cell OX40 with its ligand (OX40L), generally expressed by antigen-presenting cells, promotes T-cell proliferation and cytotoxicity [48]. Furthermore, it has been demonstrated that OX40+ follicular helper T cells aid an effective immune response through the formation of secondary lymphoid structures [49]. Interestingly, treatment with an agonistic OX40 antibody, which may recapitulate T-cell stimulation through OX40L+ myeloid cells, resulted in the systemic appearance of effector T cells expressing NK cell receptors and chemo-attractant receptors [50]. Moreover, such treatment of patients with HNSCC resulted in the enhanced recruitment of antigen-reactive T cells into the tumor [51]. These reports may be supportive of our finding that frequencies of OX40+ T cells went hand-in-hand with frequencies of more differentiated T cells, as well as CCR1, 4 and/or 5+ T cells in the blood of EBV DNA clearers. Along this line, OX40 represents a candidate receptor for future studies investigating the mechanism of action of NPC TILs, and to enhance the migration and responsiveness of effector T cells in NPC. Besides the co-stimulatory receptor OX40, we also observed a temporal increase in the frequency of CD8+ T cells expressing the co-inhibitory receptor TIM3 in plasma EBV DNA clearers. The interaction between TIM3 and its ligand Galectin9 is a reported immunosuppressive pathway enriched in recurrent NPC, and has been shown to recruit Treg cells following EBV infection [13,52]. An increase in the abundance of TIM3-expressing CD8+T cells could be a footprint of NPC-TILs as a consequence of chronic EBV infection as reported by Liu and colleagues [12]. Future studies assessing the specificity of T cells with the here-described phenotype are valuable. To this end, next steps would include assessment of the co-occurrence of the T-cell phenotype, the binding of EBV-specific peptide-multimers (by flow cytometry) and/or the enrichment of T cells with this phenotype by FACSort and the performance of single-cell analysis to retrieve the usage of EBV TCR genes.

Our study has several limitations and/or recommendations. First, the patient sample size was small and our results must be validated in larger cohorts. Inherent to the small sample size is that we could not distinguish between different treatment regimens, such as radiotherapy or concurrent chemo-radiotherapy, with or without induction therapy to better assess the differential effects of different regimens. Despite the low number of patients, we did observe some temporal changes in that group that highlight the strength of our analysis. Second, the time post-RT for evaluating the consistency of effects is short and our findings should be evaluated to confirm if they are true T-cell phenotype driving mechanisms or after-effects of RT-induced cell death. Third, paired measurements of plasma EBV DNA and chemo-attractant levels at different timepoints relative to RT in Cohort 1 were not performed due to insufficient amounts of biomaterials. Ensuring the collection of sufficient biomaterials (i.e., PBMCs and plasma) at all considered timepoints should be a prerequisite of newly planned prospective studies. Finally, our outcomes should be further investigated using immune staining on primary and recurrent NPC tumors to assess which T-cell subsets are differentially present in patients who develop recurrence and their correlation with the circulating T-cell profile. Specifically, the profiling of myeloid populations in circulating and local TME can also help elucidate mechanisms inducing T-cell evasion. The above further analyses and future studies based on our recommendations will be instrumental in generating a more complete picture of the underlying immunological profile associated with post-treatment EBV clearance and better clinical outcomes.

## 5. Conclusions

This pilot study is the first to identify RT-induced changes in the phenotypic profiles of circulating CD8+ and CD4+ T cells in NPC patients and their association with post-RT plasma EBV DNA clearance and tumor recurrence. Our findings highlight the impact of RT on the systemic T-cell profile and offer potential surrogate markers, such as CCR1-, 4- and/or 5-expressing T cells, which can be tested in clinical trials investigating immunotherapies to improve NPC patient stratification and response monitoring. Targeting these specific T-cell phenotypes also offers opportunities for developing novel and improved immunotherapies for patients with advanced NPC.

## Figures and Tables

**Figure 1 cancers-15-01887-f001:**
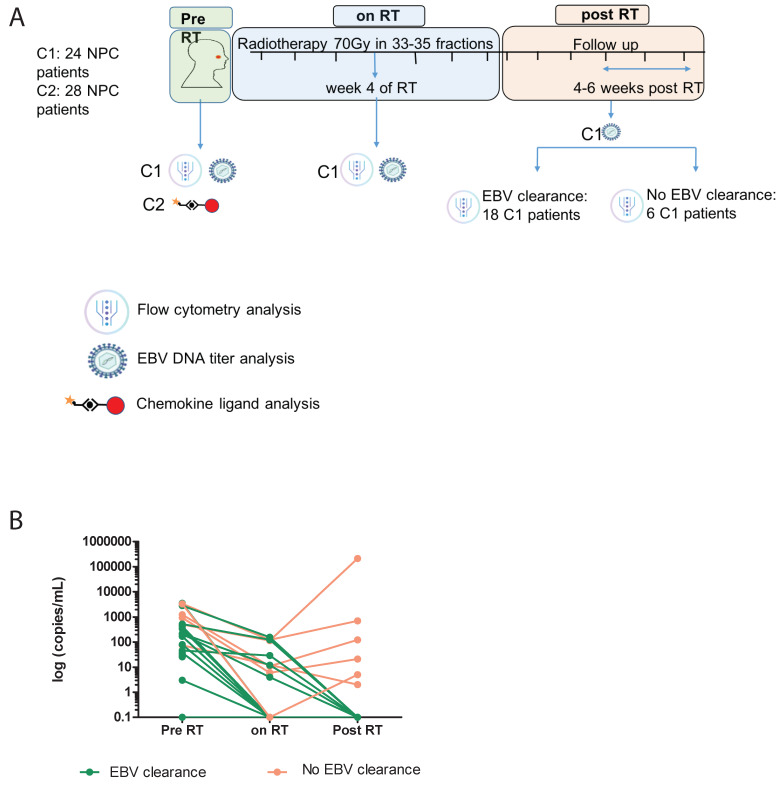
Study design of NPC treatment, collection of biomaterials and detection of plasma EBV DNA and T-cell profiling. (**A**) Schematic diagram of NPC patient Cohorts 1 and 2, timeline of radiotherapy, collection of biospecimens and readouts. Patients diagnosed with NPC received 7 weeks of radiotherapy (RT) with or without chemotherapy. Whole blood was collected to isolate PBMCs as well as plasma. The latter was used to calculate EBV DNA titers or chemokine ligand levels. Epstein–Barr virus DNA titers were measured at pre-RT, at week 4 during RT (on-RT) and between 4 and 6 weeks after RT (post-RT). Peripheral blood mononuclear cells were used for flow cytometry analysis of T-cell subsets according to >20 markers of maturation, co-signaling and chemotaxis. Plasma EBV DNA titers at post-RT were used to stratify patients as those who cleared or did not clear EBV DNA. See Materials and Methods for details. (**B**) Line plots showing plasma EBV DNA titers of NPC patients at all time points. Patients who did or did not clear plasma EBV DNA are indicated with green or red, respectively.

**Figure 2 cancers-15-01887-f002:**
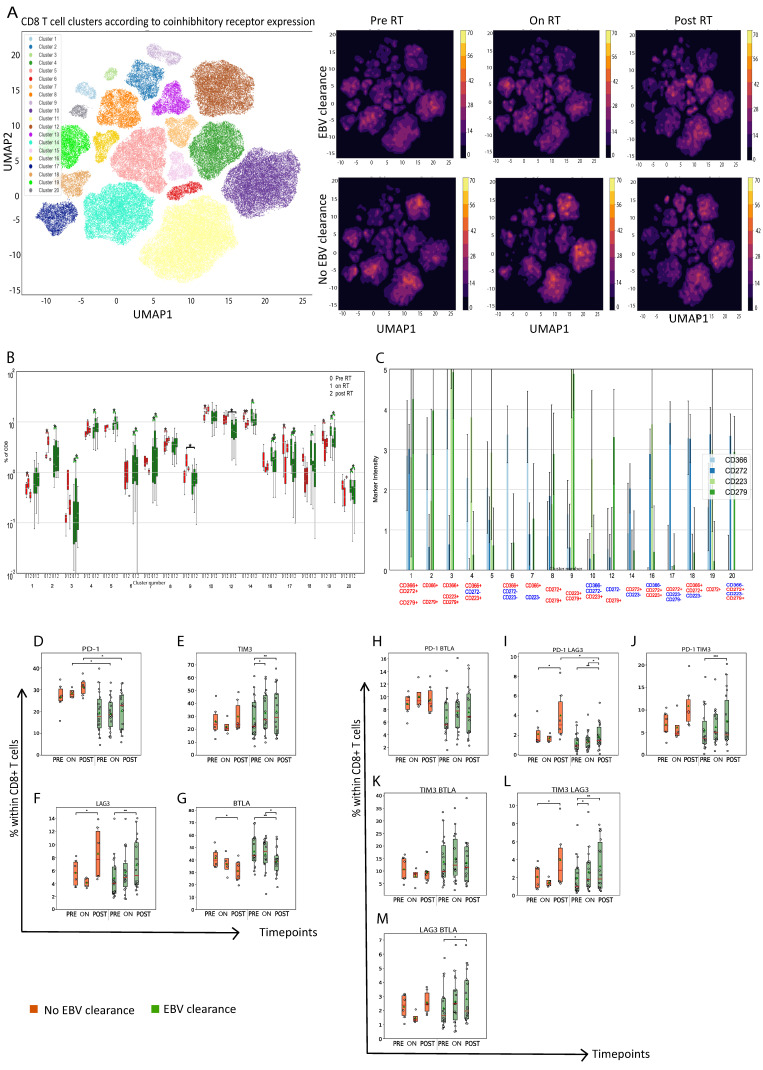
Plasma EBV DNA clearers harbor lower frequency of PD1-expressing CD8+ T cells on- and post-RT when compared to non-clearers. (**A**) A UMAP scatter representation of clusters of CD8+ T cells (co-)expressing 4 co-inhibitory receptors (24 patients, 3 timepoints) (left-hand side plot) and split according to plasma EBV DNA titers and timepoints (right-hand side plot). (**B**) Bars depicting abundance of clusters with differential abundance between clearers and non-clearers and/or between different timepoints. Statistically significant difference in abundance of individual clusters is indicated with an asterisk(s); black corresponds to inter-group comparison, and green and red correspond to intra-group comparisons. (**C**) Bars showing intensities of clusters that were differently abundant according to (**B**). Particular immune markers are displayed in red or blue in cases where they are expressed above or below the average intensity, respectively. Boxplots displaying fractions of CD8+ T cells expressing a single type (**D**–**G**) or (**H**–**M**) two different types of co-inhibitory receptors. Green: clearers (n = 18); red: non-clearers (n = 6). Pre-RT: before radiotherapy; on-RT: week 4 of radiotherapy; post-RT: 4 to 6 weeks after radiotherapy. Statistically significant differences between the patient groups were determined using the Mann–Whitney U test; differences between timepoints were determined for paired samples using the Wilcoxon signed rank test. * *p* < 0.05, ** *p* < 0.01, *** *p* < 0.001.

**Figure 3 cancers-15-01887-f003:**
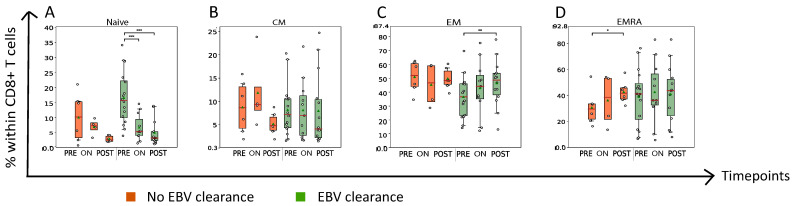
Plasma EBV DNA clearers demonstrate steep and early drop in frequency of naïve CD8+ T cells and increase in frequency of effector memory CD8+ T cells during RT. Boxplots displaying frequency of CD8+ T cells according to different stages of maturation ranging from (**A**) naive T cells (CCR7+CD45RA+), (**B**) central memory (CM) (CCR7+CD45RA-), (**C**) effector memory (EM) (CCR7-CD45RA-) and (**D**) terminal effector EMRA (CCR7-CD45RA+) T cells. Details regarding display of patient groups, time points and statistical analyses are described in legend to Figure 2. * *p* < 0.05, ** *p* < 0.01, *** *p* < 0.001.

**Figure 4 cancers-15-01887-f004:**
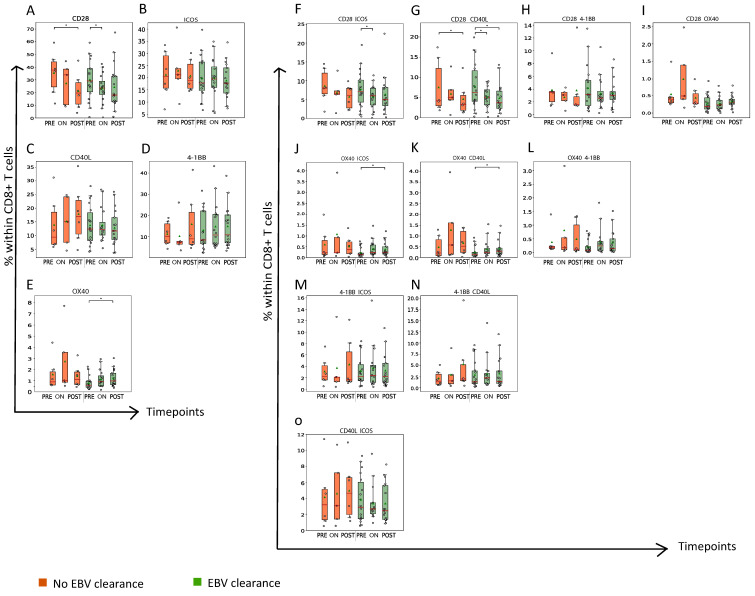
Plasma EBV DNA clearers demonstrate late increase in frequency of OX40-expressing CD8+ T cells during RT. Boxplots displaying frequency of CD8+ T cells expressing (**A**–**E**) a single type and (**F**–**O**) two different types of co-stimulatory receptors. Details regarding display of patient groups, time points and statistical analyses are described in legend to Figure 2. * *p* < 0.05.

**Figure 5 cancers-15-01887-f005:**
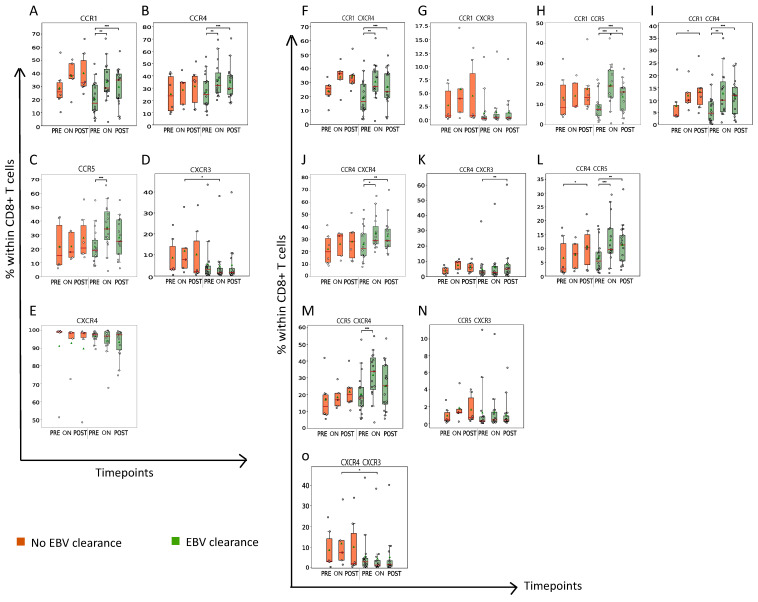
Plasma EBV DNA clearers demonstrate early increase in frequency of CCR1-, CCR4- and/or CCR5-expressing CD8+ T cells during RT. Boxplots displaying frequency of CD8+ T cells expressing (**A**–**E**) a single type or (**F**–**O**) two different types of chemo-attractant receptors. Details regarding display of patient groups, time points and statistical analyses are described in legend to Figure 2. * *p* < 0.05, ** *p* < 0.01, *** *p* < 0.001.

**Figure 6 cancers-15-01887-f006:**
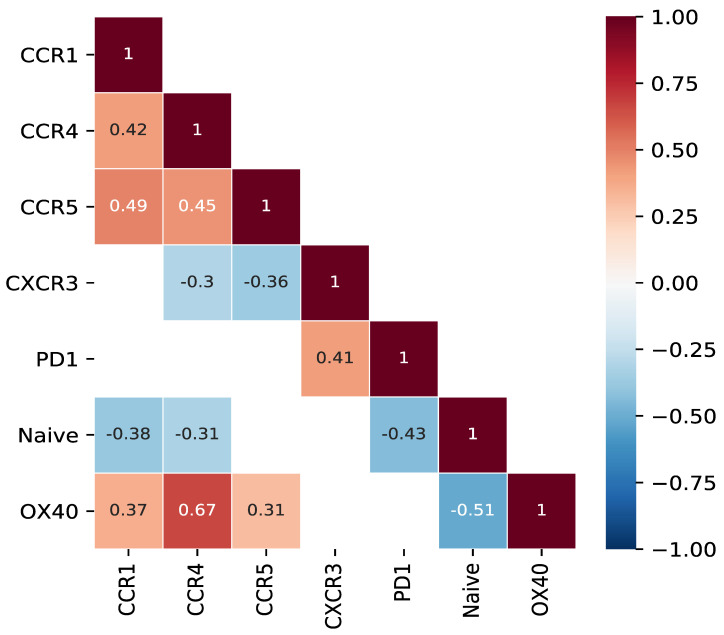
Frequency of CD8+ T cells expressing CCR1, CCR4 and/or CCR5 correlates positively with frequency of CD8+ T cells expressing OX40, and negatively with frequency of naïve CD8+ T cells. Correlation heatmap showing Spearman correlation coefficients between frequencies of CD8+ T-cell subsets expressing selected coinhibitory, co-stimulatory, chemo-attractant receptors as well as maturation markers from Figure 2, Figure 3, Figure 4 and Figure 5. Only significant correlations are shown (*p* value < 0.05).

**Figure 7 cancers-15-01887-f007:**
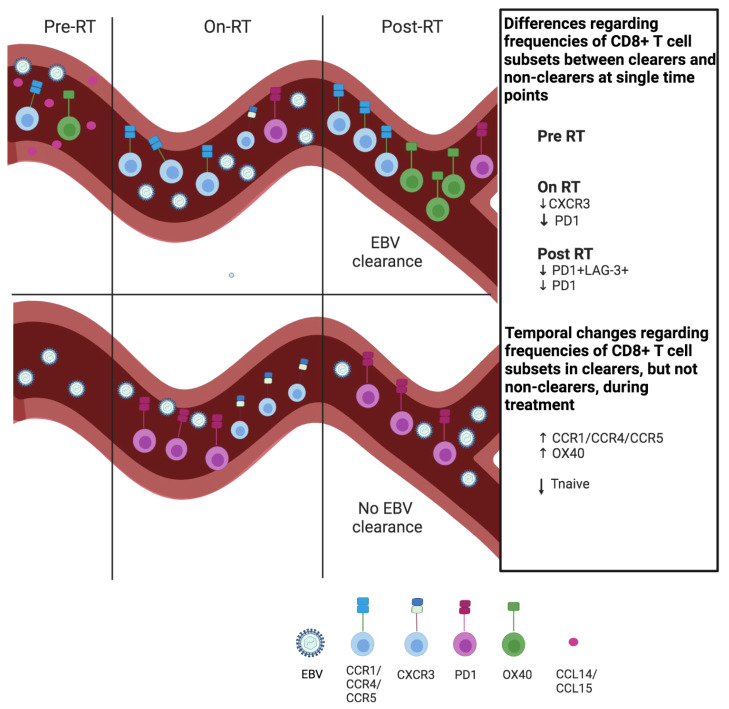
Overview of CD8+ T-cell profile in blood of NPC patients upon radiotherapy according to plasma EBV DNA clearance. Schematic overview of abundances of peripheral CD8+ T-cell subsets that characterize post-RT plasma EBV DNA clearers. First, differences are displayed at single time points in clearers versus non-clearers, and second, treatment-induced changes are displayed that occur in plasma EBV DNA clearers, but not non-clearers. See Table 2 for a short description of the T-cell markers mentioned. This table is made with bioRender.

**Table 1 cancers-15-01887-t001:** Patient characteristics of cohorts 1 and 2.

**Cohort 1**		
Age, median (range)		51 (30–67)
Sex, n		
	Male	20
	Female	4
Tumor stage, n		
	I	3
	II	3
	III	11
	IVA	7
Treatment		
	Induction chemotherapy followed by concurrent chemoradiotherapy	5
	Concurrent chemoradiotherapy	21
	Radiotherapy	3
**Cohort 2**		
Age, median (range)		55 (33–68)
Sex, n		
	Male	21
	Female	7
Tumor stage, n *		
	I	1
	II	5
	III	11
	IVA	5
Treatment		
	Induction chemotherapy followed by concurrent chemoradiotherapy	7
	Concurrent chemoradiotherapy	21
	Radiotherapy	5

* Clinical staging information of some patients in Cohort 2 is unavailable since they were treated in private care in Hong Kong.

**Table 2 cancers-15-01887-t002:** List of selected T-cell markers that show changes in blood of NPC patients upon radiotherapy according to plasma EBV DNA clearance.

Marker	Expression	Ligand	Immune Function	Therapeutic Target
PD-1 [25]	Expressed by different immune and myeloid cells	PD-L1	Inhibits TCR-CD3z and CD28 signaling via recruitment of phosphatases.	Antibodies targeting PD1 are standard of care for multiple cancer types
LAG3 [26,27]	Expressed by different immune cells	MHC-II	Inhibits early steps of TCR signaling, such as NFAT activation	Antibodies targeting LAG3 to treat multiple tumor types are currently tested in clinical trials
OX40 [28,29]	Expressed by CD4+ and CCD8+ T cells	OX40L	Promotes survival and memory generation of CD4+ T cells and suppresses regulatory T-cell activation by antagonizing TGF β signaling	Agonistic antibody-targeting OX40 resulted in enhanced recruitment of antigen-reactive T cells into HNSCC tumor in a clinical trial
CXCR3 [30]	Expressed by different immune and myeloid cells	CXCL9/10/11/13	Enhances recruitment and migration of T cells	No clinical trials yet that target CXCR3
CCR1 [31]		CCL3/4/5/8/13/14/15/23	Enhances recruitment of T cells and suppressive macrophages	No clinical trials yet that target CCR1
CCR4 [32,33]		CCL3/5/17/22	Enhances recruitment of T cells and, in the case of the CCR4-CCL22 axis, mediates suppression by regulatory T cells	CCR4 antagonist blocks CCR4-expressing regulatory T cells for treatment of T-cell leukemia
CCR5 [34,35]		CCL2/3/4/5/8/11/13/14	Enhances recruitment of T cells, and constitutes an entry receptor of HIV. Activates calcium signaling and PI3K pathway to induce survival	CCR5 antagonist repolarizes macrophages to induce an anti-tumor effect in colorectal cancer patients.

T-cell markers, as highlighted in Figure 7, are explained regarding expression, ligand, immune function and therapeutic targeting.

## Data Availability

The Appendix A have been provided with the manuscript.

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
