# Peer review of "Frequency of Peripheral CD8+ T Cells Expressing Chemo-Attractant Receptors CCR1, 4 and 5 Increases in NPC Patients with EBV Clearance upon Radiotherapy"

_cancers, 2023, doi:10.3390/cancers15061887_

Round 1

Reviewer 1 Report

This is a fascinating pilot study that outlines the potential relationship of chemokine specific T-cell subsets to clearance of EBV DNA from plasma in NPC.

The reader has several questions that I hope the authors could address

1) There is an emphasis on the OX-40 axis in the manuscript, yet very little was discussed with respect to OX-40 in NPC.  Could the authors elaborate further?

2) Were the described T-cells EBV-specific in nature? 

3) CCR4 + CD8+ T-cells have been described to associate with  IL-4, IFN-gamma, IL-2, and TNF-alpha (Kondo 2009) production.  The authors looked at Luminex panels of chemo-attractants but did not describe if they found any associations.  Did the authors look at other cytokine levels from plasma in their study and if so were there any associations seen.

4) The limitation paragraph is adequate.

5) Methods are adequately described

Author Response

Reviewer #1:
This is a fascinating pilot study that outlines the potential relationship of chemokine specific T-cell subsets to clearance of EBV DNA from plasma in NPC. The reader has several questions that I hope the authors could address.

Comment 1: There is an emphasis on the OX-40 axis in the manuscript, yet very little was discussed with respect to OX-40 in NPC. Could the authors elaborate further?

Response
The authors would like to thank R1 for requesting clarification regarding data on OX40+ T cells in the blood of NPC patients and its interpretation. We observed that NPC patients with plasma clearance of EBV DNA, but not those with no plasma clearance of EBV DNA, demonstrated an increase in the frequencies of OX40+ CD4 as well as CD8 T cells in their blood upon treatment with concurrent (chemo)radiation (see Figure 4 and supplementary figure 4 of manuscript). OX-40 is a co-stimulatory receptor that is expressed by different subsets of T cells, particularly CD4 T cells, and its expression is induced following antigen exposure.(1) Thus, the increased frequency of OX40+ T cells in the blood may be reflective of an antigen-encounter in the tumor or tumor-draining lymphoid organs, the occurrence of which may have been facilitated upon (chemo)radiation-induced immunogenic cell death of cancer cells and subsequent uptake and presentation of released antigens by myeloid cells. In fact, interaction of T cell-OX40 with its ligand (OX40L), generally expressed by antigen-presenting cells, promotes T cell proliferation and cytotoxicity.(2) Furthermore, it has been demonstrated that OX40+ follicular helper T cells aid an effective immune response through the formation of secondary lymphoid structures.(3)  Interestingly, treatment with an agonistic OX40 antibody, which may recapitulate T cell stimulation through OX40L+ myeloid cells, resulted in systemic appearance of effector T cells expressing NK cell receptors and chemo-attractant receptors.(4) Moreover, such treatment of patients with Head and neck squamous cell carcinoma (HNSCC) resulted in enhanced recruitment of antigen-reactive T cells into the tumor.(5) Taken together, these reports may be supportive of our finding that frequencies of OX40+ T cells went hand-in-hand with frequencies of more differentiated T cells, as well as CCR1, 4 and/or 5+ T cells in the blood of EBV DNA clearers. Along this line, OX40 represents a candidate receptor for future studies investigating the mechanism of action of NPC tumor infiltrating lymphocytes (TILs), and to enhance migration and responsiveness of effector T cells in NPC. We have added these details in the discussion section of the manuscript (see highlighted sections in the manuscript).

Comment 2: Were the described T-cells EBV-specific in nature? 

Response
The aim of our study was to search for a T cell phenotype that could serve as a surrogate marker for EBV DNA clearance and clinical response in NPC patients, not necessarily related to EBV specificity. However, the authors fully concur with R1 regarding the relevance of EBV specificity in relation to T cell phenotypes. In fact, our observation of enhanced frequencies of OX40 and /or TIM3+ T cells in the blood of EBV clearers are in line with a co-activation-exhaustion signature which has been reported to be a footprint of a chronic viral infection(6). Following the definition of a T cell phenotype that is related to EBV clearance (as presented in current manuscript), future studies assessing the specificity of these T cells become valuable. To this end, assessing the co-occurrence of the described T cell phenotype and the binding of EBV-specific peptide-multimers (by flow cytometry), or to enrich T cells with this phenotype by FACSort and perform single cell analysis to retrieve the usage of EBV TCR genes are highly relevant follow up analysis. Such studies are now mentioned in the Discussion section of the revised manuscript.

Comment 3: CCR4+ CD8+ T-cells have been described to associate with IL-4, IFN-gamma, IL-2, and
TNF-alpha (Kondo, 2009) production. The authors looked at Luminex panels of chemo-attractants but did
not describe if they found any associations. Did the authors look at other cytokine levels from plasma in
their study and if so were there any associations seen.

Response
In our study, particularly using the first cohort of patients, we have observed an increase in the frequency of CCR1, 4 and/or 5+ T cells in the blood of EBV DNA clearers following (chemo)radiation. To extend these findings, and assess the levels of chemo-attractants in plasma, we introduced a second cohort as plasma samples from the first cohort were insufficiently available (see Table 1 in manuscript for information on both cohorts). Of this second cohort, clinically similar to the first cohort, plasma samples were available from pre-RT timepoints, yet no plasma samples at later timepoints nor PBMCs at any timepoint were available. Even though both cohorts were highly valuable for our studies, the unfortunate scarcity of biomaterials did not allow for correlative analysis across PBMCs and plasma. Furthermore, the analysis of plasma samples from the second cohort focused on chemo-attractant ligands, particularly those that correspond to CCR1, 4 and 5, and did not include cytokines like IL2, IL4, IFN nor TNF. As R1 correctly points out, Kondo and colleagues have reported an activated phenotype of CCR4+ CD8+ T cells, as evidence by the production of inflammatory cytokines.(7) Our findings build further on this notion, as the frequencies of  CCR1, 4 and/or 5+ T cells correlated with the frequencies of T cells with an activated phenotype, as governed by the expression OX40 and T cell maturation markers. Additional analysis whether CCR1, 4 and/or 5+ T cells produce inflammatory cytokines is relevant and is recommended for future research making use of patient cohorts with more complete sets of different biomaterials. These recommendations have been mentioned in the discussion section of our manuscript. 

Comments 4, 5: The limitation paragraph is adequate, and methods are adequately described.

Response
Authors thank R1 for his/her remark.  

References to this rebuttal
1. Moran, A. E., M. Kovacsovics-Bankowski, and A. D. Weinberg. 2013. The TNFRs OX40, 4-1BB, and CD40 as targets for cancer immunotherapy. Curr. Opin. Immunol. .
2. Deng, J., S. Zhao, X. Zhang, K. Jia, H. Wang, C. Zhou, and Y. He. 2019. OX40 (CD134) and OX40 ligand, important immune checkpoints in cancer. Onco. Targets. Ther. .
3. Tahiliani, V., T. E. Hutchinson, G. Abboud, M. Croft, and S. Salek-Ardakani. 2017. OX40 Cooperates with ICOS To Amplify Follicular Th Cell Development and Germinal Center Reactions during Infection. J. Immunol. .
4. Beyrend, G., T. van der Sluis, E. van der Gracht, T. Abdelaal, S. Jochems, R. Belderbos, T. Wesselink, S. van Duikeren, F. van Haften, A. Redeker, E. Beyranvand Nejad, M. G. M. Camps, K. Franken, M. Linssen, P. Hohenstein, N. F. C. C. de Miranda, H. Mei, A. Bins, J. Haanen, J. G. Aerts, F. A. Ossendorp, and R. Arens. 2022. OX40 Agonism Enhances Efficacy of PD-L1 Checkpoint Blockade by Shifting the Cytotoxic T Cell Differentiation Spectrum. SSRN Electron. J. .
5. Duhen, R., C. Ballesteros-Merino, A. K. Frye, E. Tran, V. Rajamanickam, S. C. Chang, Y. Koguchi, C. B. Bifulco, B. Bernard, R. S. Leidner, B. D. Curti, B. A. Fox, W. J. Urba, R. B. Bell, and A. D. Weinberg. 2021. Neoadjuvant anti-OX40 (MEDI6469) therapy in patients with head and neck squamous cell carcinoma activates and expands antigen-specific tumor-infiltrating T cells. Nat. Commun. .
6. Jin, S., R. Li, M. Y. Chen, C. Yu, L. Q. Tang, Y. M. Liu, J. P. Li, Y. N. Liu, Y. L. Luo, Y. Zhao, Y. Zhang, T. L. Xia, S. X. Liu, Q. Liu, G. N. Wang, R. You, J. Y. Peng, J. Li, F. Han, J. Wang, Q. Y. Chen, L. Zhang, H. Q. Mai, B. E. Gewurz, B. Zhao, L. S. Young, Q. Zhong, F. Bai, and M. S. Zeng. 2020. Single-cell transcriptomic analysis defines the interplay between tumor cells, viral infection, and the microenvironment in nasopharyngeal carcinoma. Cell Res. .
7. Kondo, T., and M. Takiguchi. 2009. Human memory CCR4+CD8+ T cell subset has the ability to produce multiple cytokines. Int. Immunol. .

Reviewer 2 Report

The study presents some interesting observations regarding good immunomodulation post-radiotherapy among NPC patients. However, there are some points that should be discussed. 

(1) The patients in group one received several doses of chemotherapy concurrent cisplatin (40mg/m2), which was completely ignored by the authors. this point needs to be discussed, why authors didn't present a control group of patients that received only RT without chemotherapy to make the right conclusion?

(2) Another issue, the authors provided a lot of data without clear interpretation or connection. There many mechanisms and markers were presented, which belong to different pathways. the connection between those mechanisms and markers was weak. Thus, I suggest authors focus on T cell sub-population changes.  

(3) I advise authors to write study limitations and mention the missed controls and other issues. 

(4) the time post-RT for evaluating the consistency of effect is very short, it shouldn't be less than 3 months. Thus, please add this point to the limitation of this study. 

(5) The final conclusion of the study isn't clear. It shouldn't mention the all findings and markers. It should focus on the main findings. 

Author Response

The authors thank you for your contribution to the manuscript in the shape of the comments that have added value to this manuscript and made it more clear. please see the attachment for our response to your comments. 

Reviewer 3 Report

In the manuscript entitled "Frequency of peripheral CD8+ T cells expressing chemo-attractant receptors CCR1, 4 and 5 increases in NPC patients with EBV clearance upon radiotherapy" the authors did extensive study on the changes in phenotype of the circulating T cells in NPC patients undergone through RT. The authors also tried to point out potential biomarkers that can be tested for immunotherapies to improve NPC patients  stratification and response.

The manuscript is nicely drafted and the results were analysed and presented in a easy way for better understanding. 

The manuscript lacks in the introduction section. The authors could not express why they are performing this study. Moreover the authors did multicolor flowcytometric analysis to study the differnet phenotypes of T cells. The authors should mention the function of each of this markers in the result section as well so that the reader can understand the significance of those markers. 

Otherwise the results are convincing and being a pilot study, it gives a lot of information can be further studied.

Author Response

The authors thank you for your contribution to the manuscript in the shape of the comments that have added value to this manuscript and made it more clear. Please see the attachment for our response to your comments. 

Round 2

Reviewer 2 Report

The authors addressed good points. I can accept it.